# Pulmonary Adenocarcinoma In Situ and Minimally Invasive Adenocarcinomas in European Patients Have Less *KRAS* and More *EGFR* Mutations Compared to Advanced Adenocarcinomas

**DOI:** 10.3390/ijms25052959

**Published:** 2024-03-03

**Authors:** Jennie Petterson, Dyar Mustafa, Sashidar Bandaru, Ella Äng Eklund, Andreas Hallqvist, Volkan I. Sayin, Andréanne Gagné, Henrik Fagman, Levent M. Akyürek

**Affiliations:** 1Department of Clinical Pathology, Sahlgrenska University Hospital, Västra Götalandsregionen, 413 45 Gothenburg, Sweden; 2Department of Medical Chemistry and Cell Biology, Institute of Biomedicine, University of Gothenburg, Sahlgrenska Academy, 405 30 Gothenburg, Sweden; 3Sahlgrenska Center for Cancer Research, University of Gothenburg, Gothenburg, Sahlgrenska Academy, 405 30 Gothenburg, Sweden; ella.ang@gu.se (E.Ä.E.);; 4Department of Clinical Oncology, Sahlgrenska University Hospital, Västra Götalandsregionen, 413 45 Gothenburg, Sweden; 5Department of Surgery, Institute of Clinical Sciences, University of Gothenburg, 413 45 Gothenburg, Sweden; 6Wallenberg Centre for Molecular and Translational Medicine, University of Gothenburg, 413 45 Gothenburg, Sweden; 7Department of Pathology, Brigham and Women’s Hospital, Harvard Medical School, Boston, MA 02115, USA; 8Department of Laboratory Medicine, Institute of Biomedicine, University of Gothenburg, Sahlgrenska Academy, 405 30 Gothenburg, Sweden

**Keywords:** lung adenocarcinoma, *KRAS*, *EGFR*, mutation, biomarker

## Abstract

Pulmonary adenocarcinoma (ADC) is a very diverse disease, both genetically and histologically, which displays extensive intratumor heterogeneity with numerous acquired mutations. ADC is the most common type of lung cancer and is believed to arise from adenocarcinoma in situ (AIS) which then progresses to minimally invasive adenocarcinoma (MIA). In patients of European ethnicity, we analyzed genetic mutations in AIS (*n* = 10) and MIA (*n* = 18) and compared the number of genetic mutations with advanced ADC (*n* = 2419). Using next-generation sequencing, the number of different mutations detected in both AIS (87.5%) and MIA (94.5%) were higher (*p* < 0.001) than in advanced ADC (53.7%). In contrast to the high number of mutations in Kirsten rat sarcoma virus gene (*KRAS*) in advanced ADC (34.6%), there was only one case of AIS with *KRAS* G12C mutation (3.5%; *p* < 0.001) and no cases of MIA with *KRAS* mutation (*p* < 0.001). In contrast to the modest prevalence of epidermal growth factor receptor (*EGFR*) mutations in advanced ADC (15.0%), the fraction of *EGFR* mutant cases was higher in both in AIS (22.2%) and MIA (59.5%; *p* < 0.001). The *EGFR* exon 19 deletion mutation was more common in both MIA (50%; *n* = 6/12) and ADC (41%; *n* = 149/363), whereas p.L858R was more prevalent in AIS (75%; *n* = 3/4). In contrast to pulmonary advanced ADC, *KRAS* driver mutations are less common, whereas mutations in *EGFR* are more common, in detectable AIS and MIA.

## 1. Introduction

Recent discoveries have increased our knowledge of the molecular genetic pathways in non-small cell lung cancer (NSCLC) cells. Additionally, NSCLC is considered a biologically heterogeneous carcinoma because the initiation and/or progression of NSCLC are driven by multiple genetic mutations [1]. Mutations causing the oncogenic alterations in these pathways have been defined, and genetic tests for them have become available clinically. As continuously provided by the National Comprehensive Cancer Network (NCCN), new inhibitors targeting these aberrant pathways have been developed with better outcomes, higher specificity, and fewer cytotoxic effects compared to traditional chemotherapies (https://www.nccn.org/guidelines/ accessed on 20 February 2024).

Advanced adenocarcinoma (ADC) is the most common type of lung cancer, representing about 40% of all NSCLC tumors that are initiated by oncogenic activation and appear to be derived from targeted cells located in the bronchioalveolar duct junction, a region where the airways terminate and form alveoli [2]. Close examination of these early lung tumor lesions have revealed the presence of a rare cell population shown to co-express the Clara cell marker [3]. Recent observations even point to the contribution of bronchioalveolar stem cells [4]. Histomorphologically, adenocarcinoma in situ (AIS) and minimally invasive adenocarcinoma (MIA) are the early stages of pulmonary ADC. AIS is defined as tumors with a maximum diameter of 3 cm consisting of neoplastic pneumocytes with a purely lepidic growth pattern, without invasion. Conversely, primary MIAs measure ≤ 3 cm in greatest dimension and exhibit ≤ 0.5 cm of either stromal invasion or non-lepidic growth pattern [5].

In NSCLC, targeted therapies against somatic gene mutations as well as rearrangements in Kirsten rat sarcoma virus gene (*KRAS*), epidermal growth factor receptor (*EGFR*), anaplastic lymphoma kinase (*ALK*), ROS proto-oncogene 1 (*ROS1*), B-Raf proto-oncogene (*BRAF*), neurotrophic tyrosine receptor kinase (*NTRK*), mesenchymal–epithelial transition kinase (*MET*), and receptor tyrosine kinase (*RET*) have been designed and approved by the Food and Drug Administration and the European Medicines Agency and provided by the NCCN guidelines. *KRAS* is the most frequently mutated proto-oncogene in NSCLC (20–40%) and its prevalence is higher in Western than Asian populations (26% versus 11%) [6]. The most frequent *KRAS* mutations occur in codons 12 and 13, with the most common mutations being p.G12C, p.G12V, and p.G12D [6]. The second most common driver mutations in NSCLC occur in *EGFR*, a gene that encodes for the epidermal growth factor receptor [7]. These *EGFR* mutations can be classified as common or rare and vary in prevalence with racial background. Up to 38% of NSCLC patients in Asia have *EGFR* mutations, while they are observed in only 24% of North and South American patients, and 14% of European patients [8]. The most common *EGFR* mutations are deletions within exon 19 and the single amino acid substitution p.L858R in exon 21. These mutations, which account for about 85% of observed *EGFR* mutations in NSCLC, lead to constitutively active *EGFR* signaling that promotes tumor growth [7]. 

Molecular analysis of AIS or MIA is not commonly performed as surgical lobectomy and also sublobar resection is considered curative without complementary neoadjuvant chemo- or immunotherapy with respect to disease-free survival [9]. Thus, comprehensive insight into genetic events at these early stages is lacking. It is unknown whether genetic somatic mutations and/or rearrangements occur sequentially or simultaneously during initiation (AIS and MIA) and progression of NSCLC. In this study, we investigated genetic alterations in AIS and MIA compared to ADC in a European population.

## 2. Results

### 2.1. Histological Findings

A histopathological review in the clinical biobank database resulted in the identification of 10 cases as purely AIS (Figure 1A,B), 18 cases as MIA (Figure 1C), and 2419 cases as ADC (Figure 2 and Figure 3). All of these early adenocarcinomas had no other foci of current or previous advanced pulmonary ADC.

All patients with AIS were of European ethnicity (100%; *n* = 10/10), whereas only one patient with MIA was of Asian ethnicity (5.5%; *n* = 1/18). There was no preferential localization in the lung among the AIS cases, but most MIA cases were detected in the right upper lobe (39%; *n* = 7/18). 

### 2.2. Genetic Findings

The total number of cases without detected mutations in this study were significantly lower (*p* < 0.001) for both AIS (12.5%) and MIA (5.5%) compared to cases with ADC (46.3%), indicating mutation-rich genetic alterations in both AIS and MIA (Figure 2). The number of *KRAS* mutations in both AIS (3.5%; *n* = 1/10) and MIA (0%; *n* = 0/18) were significantly lower (*p* < 0.001) than the number of *KRAS* mutations in ADC (34.6%; *n* = 837/2419; Figure 2). *EGFR* mutations were detected in 4/10 of patients with AIS (22.2%), in 12/18 of patients with MIA (59.5%), and in 363/2419 of patients with ADC (15.0%; Figure 2 and Figure 3). The incidence of *EGFR* mutations in MIA was significantly higher than the number of *EGFR* mutations in ADC (*p* < 0.001, Figure 2). The most common type of *EGFR* mutation in AIS was p.L858R (75%; *n* = 3/4), and exon 19 deletions (50%; *n* = 6/12) in MIA (Figure 3). In patients with advanced ADC, small prevalent variations in *KRAS* (30–38%), *EGFR* (13.3–16.5%), and *BRAF* (3.0–5.0%) mutations, and *ALK* rearrangements (2.0–2.9%) were detected during the study period. The most common types of *KRAS* mutations in ADC were p.G12C (36–45%) and p.G12D (12–17%). The most common types of *EGFR* mutations in ADC were deletions in exon 19 (32–53%) and p.L858R (29–42%; Figure 3). Co-mutation in *KRAS* and *EGFR* was not detected in any cases with AIS, MIA, or ADC included in this study (*n* = 2447).

Regarding other genetic alterations, rearrangement in *ALK* was detected in one case of MIA (6%; *n* = 1/18) and in no AIS cases. A mutation in *BRAF* was observed in one case of both AIS (10%; *n* = 1/10) and MIA (6%; *n* = 1/18). Mutations in *ERBB2* were detected in one case of MIA (6%; *n* = 1/18) and in none of the AIS cases. Similarly, translocation in *RET* was observed in one case of MIA (6%; *n* = 1/18) and in no AIS cases. In advanced ADC, a relatively small variation in the prevalence of *ALK* rearrangements (2.0–2.9%), *BRAF* mutations (3.0–5.0%), *ERBB2* mutations (14.0–20.0%), and *RET* translocations (0.5–1.0%) was detected during the study period.

Regarding smoking history, 20% of patients with AIS (*n* = 2/10) and 72% of patients with MIA (*n* = 13/18) had never smoked (Table 1). The majority of AIS and MIA cases were detected during radiological follow-up examinations of previous cancers other than lung cancer. Thus, more than half of patients with AIS (60%) or MIA (60%) had a history of extrapulmonary tumors, including melanomas (*n* = 2/28), basal cell cancer in the skin (*n* = 1/28), and mammary (*n* = 6/28), colon (*n* = 1/28), thyroid (*n* = 1/28), prostate (*n* = 1/18), or hematological (*n* = 2/28) malignancies. Most cases were female patients for both AIS (90%; *n* = 9/10) and MIA (78%; *n* = 14/18; Table 1). The mean age of female patients was 71.1 ± 3.1 and 67.7 ± 2.4 years for AIS and MIA, respectively, whereas male patients were 66.0 ± 0.0 and 69.0 ± 4.4 years, respectively.

## 3. Discussion 

In the present study, we show a higher number of mutations and a difference in mutational profiles for both AIS and MIA early cancers compared to advanced pulmonary ADC. Despite mutations in *KRAS* being one of the most common mutations in invasive pulmonary ADC [10], we detected no *KRAS* mutations in AIS cases and a very low percentage of *KRAS* mutations in MIA cases. Stepwise development of these early pulmonary lesions is supported by the clinicopathological and molecular characteristics of these tumors [11]. However, the consensus regarding the mutation rates and particular types that drive molecular pathogenesis and progression in these early steps are still missing [11]. These observations raise questions as to whether AIS, MIA, and advanced ADC are driven by different somatic mutations and/or if AIS and MIA are not precursors of ADC, but rather different entities. In addition, no information is available describing a possible sequential order of mutations causing oncogenic behavior in ordinary pulmonary epithelial cells. Interestingly, *KRAS* mutations are not detected in early colon adenomas, and are instead detected after *APC* mutations in late adenomas during the development of colorectal carcinomas [12]. Indeed, the percentage of mutations in *KRAS* is significantly associated with higher TNM stage [13]. Conversely, *KRAS* mutations are detected early in pancreatic ductal adenocarcinomas and pancreatic intraepithelial neoplasia (*PanIN*) mutations in more progressed lesions are definitive precursors to the invasive disease [14]. Furthermore, a distinct order of mutational events has also been reported in hematological malignancies [15]. Thus, differentiation from normal lung epithelium to advanced ADC may be accompanied by the accumulation of mutations in several key genes that contribute to the maintenance of healthy pulmonary epithelia, the regulation of cell cycle control, and the inhibition of apoptosis. Pulmonary ADC may evolve because of multiple somatic lesions, with competition between subclones and sequential subclonal evolution. 

In East Asian patients, *EGFR* mutations have been commonly reported in both pre-invasive lesions including AIS and MIA (42%) and advanced ADC (65%) [16]. Similarly, in Japanese patients, *EGFR* mutations have been reported even more commonly in both pre-invasive lesions including AIS (85%) and MIA (83%) as well as in all ADC (54%) [17]. Our NGS data report a high prevalence of *EGFR* mutations in European patients with either pulmonary AIS or MIA, but a low prevalence in pulmonary advanced ADC. Using PCR and immunohistochemical analysis, a similarly high prevalence of *EGFR* mutations has been reported in both AIS (27%) and MIA (43%) [18]. In line with our observations, specific *EGFR* mutations in L858R and exon 19 deletions have been detected more commonly in AIS and MIA, respectively [18]. *EGFR* mutations may initiate progression of pulmonary ADC slower than what *KRAS* mutations cause, allowing for radiological detection of AIS or MIA with *EGFR* mutations at a higher rate before progression to pulmonary advanced ADC. In fact, we detected the majority of AIS or MIA cases in this study incidentally during either cardiovascular screening or metastatic tumor follow-ups by thorax CT-imaging. By contrast, the power of *KRAS* mutations to drive oncogenesis may be strong enough to rapidly initiate and progress pulmonary ADC, leaving practically no time frame to radiologically detect them at early phases. Indeed, *KRAS* mutations are associated with shorter survival and an increased rate of death among NSCLC patients compared to those with *EGFR* mutations [10]. In another study, *EGFR* mutations were associated with better progression-free survival and overall survival, while *KRAS* mutations were associated with worse survival in NSCLC patients receiving erlotinib, a first-generation tyrosine kinase inhibitor (TKI) [19]. *KRAS* and *EGFR* driver mutations in lung cancer differentially affect both drug transporter expression and cisplatin-induced WNT signaling, impacting responsiveness of the tumor to treatments [20]. Another aspect is that strong differences in gene expression patterns are observed between smokers and those who have never smoked in lung cancer. In this study, 46.5% of patients with early adenocarcinomas (AIS and MIA) had a smoking history. Doubling smoking pack-years has been associated with increased *KRAS*^G12C^ and less frequent *EGFR*^del19^ and *EGFR*^L858R^ mutations, whereas doubling smoking-free months has been associated with more frequent *EGFR*^L858R^ [21]. 

Recent studies have revealed considerable phenotypic diversity downstream of tumor-initiating oncogenes and also co-occurring genomic alterations, particularly in tumor-suppressive genes [22]. Phosphatidylinositol 3-kinase (*PIK3CA*) mutations frequently co-exist with *EGFR* or *KRAS* mutations in NSCLC and suggest poor prognosis in the *EGFR/KRAS* subgroup [23]. *EGFR*-driven NSCLC cases have been reported with co-mutations including Bcl-2-binding component 3 (*TP53*), *PIK3CA*, catenin-ß1 (*CTNNB1*), and retinoblastoma 1 (*RB1*) [24]. Co-mutation in *KRAS* and Kelch-like ECH-associated protein 1 (*KEAP1*)/nuclear factor erythroid 2-related factor 2 (*NFE2L2*) in NSCLC is an independent prognostic factor, predicting shorter survival, duration of response to initial platinum-based chemotherapy, and survival from the start of immune therapy [25]. Thus, NSCLC patients with concurrent *KRAS* and the above-mentioned mutations seem to have a clinically distinct behavior. In this study including more than 2400 cases, no pulmonary cancer harbored co-mutations in *KRAS* and *EGFR*. Nevertheless, contrary to this original notion of mutual exclusivity of *EGFR* and *KRAS* mutations, NSCLC patients have been recently reported to harbor *EGFR* mutations along with those in *KRAS* [26,27]. In one report, *EGFR*-mutated pulmonary ADC patients displayed a high rate of co-mutations in general (51%; *n* = 20/39), with about 16% of co-mutations in *KRAS* [28]. Interestingly, the presence of concomitant *EGFR* mutations affects cultured *KRAS*-mutated NSCLC cells response to TKI, implying that *EGFR*-TKI may represent an effective treatment option against NSCLC with *EGFR/KRAS* co-mutation [29]. These scenarios of co-mutations strengthen the need for the utilization of NGS as a powerful tool for not only providing information regarding potential actionable mutations, but also determining the prognostic and predictive value of various co-mutations in clinical settings.

A larger dataset analyzing AIS and MIA tumors will allow us to build more robust models which can deal with the heterogeneity of pulmonary ADC and could also allow us to comprehensively evaluate if these tumors indeed develop stepwise to ADC. Such a dataset would also enable us to study a chronological gene mutation profile and stratify the progression of carcinogenesis by different mutations. In addition, *KRAS* and *EGFR* mutation statuses should be evaluated simultaneously when assessing the risk of recurrence [30]. Nevertheless, among patients with NSCLC, there is significant variability in the prevalence of targetable genomic alterations according to genetic ancestry, histology, and smoking history [31]. In summary, differences in tumor stage, clinical complications, and response to targeted therapy may be dictated by the nature of mutations and/or the presence of co-mutations in pulmonary ADC. Additionally, the consequences of mutations in NSCLC cells need to be explored by mechanistic studies. Our observations represent the first demonstration of a different mutational profile between AIS and MIA versus ADC that may point to genetic differences uniquely observed in atypical epithelial cell behavior during the initiation of AIS and MIA compared to further clonal evolution in advanced ADC.

## 4. Material and Methods

### 4.1. Patient Material 

Consecutive formalin-fixed paraffin-embedded material of pulmonary AIS (*n* = 10), MIA (*n* = 18), and ADC (*n* = 2419) tumors were obtained through surgical resections (sublobar wedge resections or lobectomy), collected at the Department of Clinical Pathology, Sahlgrenska University Hospital (Gothenburg, Sweden), between 2017 and 2022. Clinical routine analyses were performed on Hematoxylin and Eosin-stained sections. Informed consent was approved by the Regional Ethics Review Board in Gothenburg, Sweden (permit number 953/18). Relevant patient characteristics were obtained from clinical data stored at the Department of Oncology, Sahlgrenska University Hospital (Gothenburg, Sweden). All patients with AIS had European ethnicity, whereas only one patient with MIA (5.56%) had Asian ethnicity. 

### 4.2. Histomorphological Review

Biobank material stored at the Department of Clinical Pathology, Sahlgrenska University Hospital (Gothenburg, Sweden), recorded in SymPathy Software v5.13.1.2, was searched for AIS, MIA, and ADC cases according to topographical and medical diagnostic codes. WHO classifications regarding growth pattern, tumor size, and size of invasion of adenocarcinoma [5] were used to diagnose either AIS or MIA by consultant thoracic pathologists to ensure inclusion of cases in diagnostic groups of this study. Regarding the group of ADC, NSCLCs without squamous differentiation were also included. 

### 4.3. Next-Generation Sequencing

The Oncomine Focus Assay (ThermoFisher Scientific, Waltham, MA, USA), a targeted, multi-biomarker assay that enables detection of hotspots, single nucleotide polymorphisms, indels, copy number variations, and gene fusions from DNA and RNA in a single workflow, was used and analyzed by Ion Torrent next-generation sequencing (NGS) systems (ThermoFisher Scientific) as previously described [32]. This analysis covers variants across 52 major genes with frequent alterations in NSCLC.

### 4.4. Statistical Analyses

Statistical analyses to compare the AIS, MIA, and ADC groups were performed using ANOVA and the Tukey–Kramer modification of the Tukey test. Comparisons between AIS versus ADC and MIA versus ADC groups were evaluated by Student’s *t*-test using GraphPad Prism 6 (GraphPad Software v10.2.0, La Jolla, CA, USA). All results were reported as the mean ± SEM. A *p* value < 0.05 was considered significant.

## Figures and Tables

**Figure 1 ijms-25-02959-f001:**
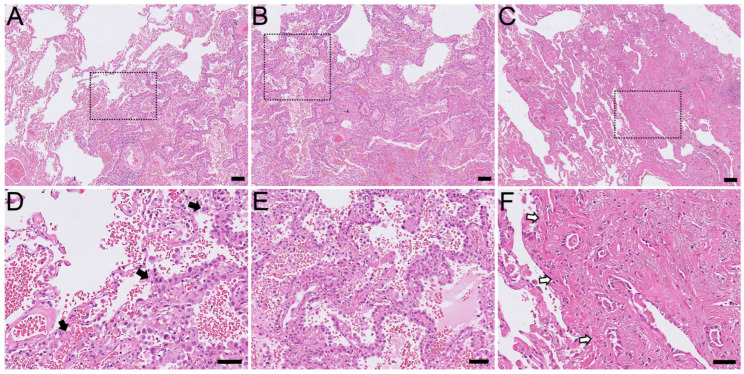
Histomorphological pictures of early pulmonary adenocarcinomas. (**A**) Adenocarcinoma in situ. (**B**) Pure lepidic growth pattern of adenocarcinoma in situ without sign of stromal invasion. (**C**) Minimally invasive adenocarcinoma. (**D**) Inset (black dotted frame) of (**A**): note lepidic growth pattern (**right**) toward uninvolved pulmonary parenchyma (**left**) as pointed to by black arrow heads. (**E**) Inset of (**B**). (**F**) Inset of (**C**): invaded atypical cells that form acinar structures are seen in invasion front as pointed to by white arrow heads. Hematoxylin and Eosin stain. Scale bars represent 100 µm in (**A**–**C**) and 50 µm in (**D**–**F**).

**Figure 2 ijms-25-02959-f002:**
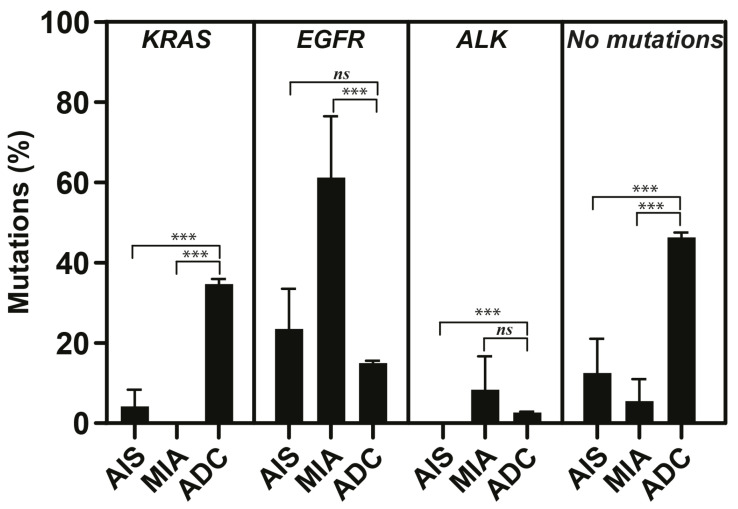
Percentages of *KRAS* and *EGFR* mutations and rearrangements in *ALK* in adenocarcinoma in situ (AIS), minimally invasive adenocarcinoma (MIA), and advanced adenocarcinoma (ADC). Percentage of cases without detected mutations using Oncomine Focus Assay were also included. Mean ± SEM values. *** *p* < 0.001. *ns:* not significant.

**Figure 3 ijms-25-02959-f003:**
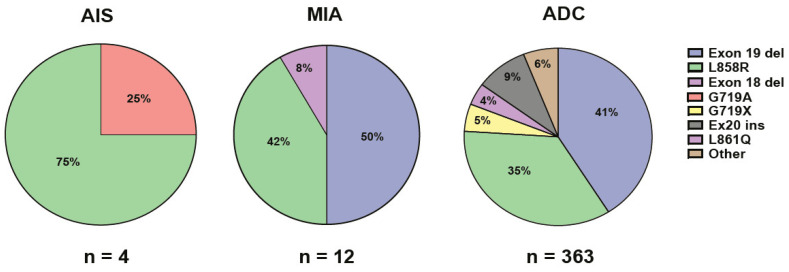
Percentages of different types of *EGFR* mutations in adenocarcinoma in situ (AIS), minimally invasive adenocarcinoma (MIA), and advanced adenocarcinoma (ADC).

**Table 1 ijms-25-02959-t001:** Patient characteristics.

	AIS + MIA (*n* = 28)	AIS (*n* = 10)	MIA (*n* = 18)
**Gender**			
Men	17.0% (5)	10.0% (1)	22.0% (4)
Women	83.0% (23)	90.0% (9)	78.0% (14)
**Age (year)**			
Men	68.0 ± 3.4	66.0 ± 0.0	69.0 ± 4.4
Women	69.0 ± 2.0	71.1 ± 3.1	67.7 ± 2.4
**Mutation**			
*EGFR*	57.1% (16)	40.0% (4)	66.0% (12)
*KRAS*	3.5% (1)	10.0% (1)	0.0% (0)
**Tumor size**			
<10 mm	32.0% (9)	40.0% (4)	28.0% (5)
10–19 mm	54.0% (15)	40.0% (4)	61.0% (11)
20–30 mm	14.0% (4)	20.0% (2)	11.0% (2)
**Lobe**			
Left upper	24.0% (7)	20.0% (2)	28.0% (5)
Left lower	18.0% (5)	30.0% (3)	11.0% (2)
Right upper	36.0% (10)	30.0% (3)	40.0% (7)
Right middle	4.0% (1)	0.0% (0)	6.0% (1)
Right lower	18.0% (5)	20.0% (2)	17.0% (3)
**Smoking history**			
Active	3.5% (1)	0.0% (0)	6.0% (1)
Prior	43.0% (12)	80.0% (8)	22.0% (4)
Never	53.5% (15)	20.0% (2)	72.0% (13)

AIS: adenocarcinoma in situ; MIA: minimally invasive adenocarcinoma. Total number of cases are given in parenthesis.

## Data Availability

Data from this study are available upon reasonable request.

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
