# Peer review of "Pulmonary Adenocarcinoma In Situ and Minimally Invasive Adenocarcinomas in European Patients Have Less KRAS and More EGFR Mutations Compared to Advanced Adenocarcinomas"

_ijms, 2024, doi:10.3390/ijms25052959_

Round 1

Reviewer 1 Report

Comments and Suggestions for Authors

Petterson et al. compared the incidence of EGFR and KRAS mutation among AIS, MIA and invasive adenocarcinoma of lung.

There is little new result in this study.

Numerous studies have showed EGFR and KRAS mutation are likely to occur in cases with or without lepidic growth, respectively.

Through the manuscript, the aim and the clinical benefit of the study is obscure. There are no prognostic data. The results does to seem to be useful in making therapeutic or diagnostic strategy.

Generally, it is not easy to follow the growing process of tumors. AIS or MIA is temporary status of tumor growth, does not reflect growth speed.

If authors have obtained data about the change of tumor growth and genetic status, please show that. Or please analyze the relationship between histological pattern and genetic status in invasive cases (for example, including lepidic component or not).

Figures include some difficulty to look. Please improve the quality.

Authors must not ignore champion data about EGFR and KRAS.

KRAS mutation can be confirmed with cases with good prognosis, and EGFR mutation can be related to aggressive tumor growth (1-3).

1. Genetic features of pulmonary adenocarcinoma presenting with ground-glass nodules: The differences between nodules with and without growth. Ann Oncol 2015; 26: 156–61.

2. Distribution and prognostic impact of EGFR and KRAS 376 mutations according to histological subtype and tumor invasion status in pTis-3N0M0 lung adenocarcinoma. BMC Cancer 2023, 377 23, 248

3. Genomic and evolutionary classification of lung cancer in never smokers. Nat Genet. 2021;53:1348–59.

These data should be included in discussion section.

Comments on the Quality of English Language

The quality is English is acceptable.

Reviewer 2 Report

Comments and Suggestions for Authors

This report shows that EGFR mutations are more common in MIA and AIS, and KRAS mutations are less common.

This is an important report because detailed genetic mutation searches are rarely performed in MIA and AIS.

I fully agree with the authors' results and discussion.

As stated in the discussion(line224-226), lung cancers with KRAS mutations progress quickly and are difficult to detect as AIS or MIA on routine imaging studies.

Slow progressing tumors are typically surgically resected after a while with a trend toward enlargement. If additional analysis is available, information on doubling time in AIS and MIA would be helpful to better visualize the clinical picture.

Minor comment

Line. 99-100 Informed consent was obtained from,,,”

Is it not correct to state that informed consent was obtained? Wouldn't it be more appropriate to state that approval was obtained from the Ethics Review Committee?

Reviewer 3 Report

Comments and Suggestions for Authors

Revision of the manuscript entitled “Pulmonary Adenocarcinoma In Situ and Minimally Invasive 2 Adenocarcinomas in European Patients Have Less KRAS and 3 More EGFR Mutations Compared to Advanced Adenocarcinomas”, by Petterson J et al. (ijms-2838602).

I thank the authors, Pettersen et al. for the opportunity to critically read and review their manuscript. The topic is certainly very timely and interesting. Indeed, the analysis of mutational status in lung cancer is full of interesting papers but few focus on early disease states, so this study certainly offers important insights for the scientific community.

However, I would have some comments, suggestions and questions to better clarify certain concepts expressed by the authors, hoping to improve the readability and understanding of the manuscript.

The case history analysed in this study is not particularly extensive: have the authors considered expanding it? It would give more support to their results. Usually, NGS studies are carried out on larger numbers. Did they consider comparing the data obtained on liquid biopsy?

Lines 54-61: The authors mention advanced adenocarcinoma (ADC), however by mentioning AIS and MIA they describe them as advanced, which they are not. I would suggest the authors either explain better or remove the part about describing advanced adenocarcinoma.

Lines 94-96: Did the authors use ADC as controls? In this case, I don't think these patients are suitable as controls since they have an advanced phenotype, I guess the authors might perform the comparison between the different phenotypes. Please explain.

Section 2.1: Why not to use liquid biopsy?

Line 102-103: In my opinion, due to what the authors stated in the manuscript, for this type of analysis (genetic study) all the patients should have the same ethnicity. Otherwise, they might include non-caucasian ethnicity to make a comparison.

As a general suggestion, I would create subsections in the Results section, describing histological findings and genetic findings.

Line 128: Again, it is not clear to me why the authors describe ADC as early adenocarcinoma. I would suggest clarification because it can be a source of misunderstanding.

General question: Did the authors think about a confirmation of these findings also on a liquid biopsy? It is well known that tissue biopsy is the gold standard and it is more sensitive than liquid biopsy, but the latter is gaining increasing importance, mainly due to its low invasiveness and repeatability.

I would suggest moving Figure 3 next to the text describing mutational characterisation.

Lines 146-148: How do the authors comment on this finding?

Lines 148-157: I would suggest that the authors put this part at the end of Results (i.e. dividing the Results into subsections, I would say in the last one) for a clearer characterisation of tumour biopsies. This would maintain continuity between the histological and genetic characterisation, then conclude with a more generic part related to other factors (smoking habits, other tumours, etc.).

I would suggest moving Table 1 to the Patients section (section 2.1) or the beginning of the Results.

Line 188-189: Please put a Reference after the sentence "Despite mutations... invasive pulmonary ADC".

Lines 193-195: There is ample evidence that adenocarcinoma is associated with several mutations in several genes (EGFR mainly, KRAS, BRAS, BRAF, RET, etc.), which may act as both disease and progression promoters, especially in the advanced state of NSCLC. Less known is the state of the art for early disease states, where genetic analysis yields fewer results.

Lines 210-215: The authors can not compare different ethnicities. What about literature data on the European population?

Lines 222-224: The sentence "In fact, we ... by CT-thorax imaging" should be avoided.

Lines 233-238: How do the authors comment on the findings on differences in smoking habits? The authors only report data in the literature, do they know of any predictive role or other associations between smoking habit and mutational status?

Lines 241-243: The following sentence is not clear to me " Phospatidylinositol .... wild-type subgroup" shouldn't it be the opposite? I mean, if I have correctly understood, the coexistence of PIK3CA and EGFR/KRAS mutation in NSCLC should give a poor prognosis in mutated patients not in EGFR/KRAS wt patients. Please explain.

Lines 250-251: Did the authors verify? No data are reported.

Line 259: NGS is certainly a powerful approach for mutational screening, although sensitivity is comparable with PCR-based techniques. However, I would not be so trenchant in favouring NGS, as in my opinion it should be considered a combination of methodological approaches (e.g. NGS and PCR-based techniques). See also the paper PMID 38116291.

Comments on the Quality of English Language

Minor revision of the Language.

Round 2

Reviewer 1 Report

Comments and Suggestions for Authors

Reviewer does not feel honesty in author's comments.

They just repeat my comments in some parts and far from persuasive in other parts.

No revision is suggested after reviewer 1' s comments.

The draft does not seem improved.

Comments on the Quality of English Language

The quality is English is acceptable.
